# Bipolar Disorder and Gaming Disorder—Compatible or Incompatible Diagnoses?

**DOI:** 10.3390/jcm12196251

**Published:** 2023-09-28

**Authors:** Georgios Floros, Ioanna Mylona

**Affiliations:** 12nd Department of Psychiatry, Aristotle University of Thessaloniki, 54624 Thessaloniki, Greece; 2Department of Ophthalmology, General Hospital of Serres, 62100 Serres, Greece; milona_ioanna@windowslive.com

**Keywords:** gaming disorder, bipolar disorder, manic episode

## Abstract

Gaming Disorder (GD) is one of the latest additions in the psychiatric taxonomy, following its official inclusion in the latest revision of the International Classification of Diseases (ICD-11). This narrative review examines the rationale of an exclusion criterion for the diagnosis of GD, that of disordered gaming being limited exclusively during an episode of elevated mood in bipolar disorder. The history of the formulation of diagnostic criteria for the disorder and all relevant published studies are examined critically, and conclusions are drawn as to the potential validity and usability of the exclusion criterion. Suggestions are offered for future research to elucidate not only the relevance of the exclusion criterion but also the differential diagnosis of GD with pathological gambling (PG).

## 1. Introduction

The recent 11th revision of the International Classification of Diseases (ICD-11) [1] has officially introduced the diagnosis of “Gaming Disorder” (GD) into the psychiatric taxonomy; this novel disorder is placed under the umbrella term of a “behavioral addiction” along with gambling disorder (PG), highlighting supposed similarities in the underlying processes and course. This narrative review will attempt to elucidate the origins of one common exclusion criterion for GD and PG, that of gaming or gambling during an episode of elated mood, by examining the history of its inclusion into the respective diagnoses, and the evidence base for and against it with regards to GD. All published research on GD in bipolar patients will be presented in an effort to determine whether there is actual evidence for, or against, excessive gaming behavior exclusively during an episode of elevated mood in bipolar patients, and whether any research was aimed at bipolar patients who demonstrate disordered gaming.

PG and GD share, in the ICD-11, a large number of common denominators, starting with their definition; PG and GD relate to a specific pattern of gambling or gaming that is essentially identical: it is characterized by impaired control, increased priority over other activities, and escalation despite negative consequences. This pattern may be continuous or episodical and recurrent. It results in marked distress or significant functional impairment, while it typically requires a duration of twelve months to be diagnosed. The only major difference mentioned in the ICD-11 as delineating the two disorders is that PG relates to striving for “obtaining something of greater value” [1].

This similarity in perception of the two disorders has a long history, starting back in 1997 [2] when Dr. Ivan K. Goldberg, a psychiatrist, attempted to ridicule the, then novel, diagnostic criteria for PG in the third version of the Diagnostic and Statistical Manual of Mental Disorders (DSM-III) [3] by applying them onto a fictional entity, that of “Internet Addiction” (IAD). In this manner, Dr. Goldberg hoped to highlight what he considered to be a fallacy: limiting the scope of a disorder by imposing a set of fixed diagnostic criteria contrary to a diagnosis based on the application of theory. That was the first ever mention of a tendency to abuse online activities, and, much to the astonishment of its creator, the term not only persisted but flourished, with colleagues stepping up to seek help for what they themselves considered as addictive use of the internet.

Tracking the evolution of the perception of PG as a psychiatric disorder is useful in understanding today’s state regarding behavioral addictions since PG is considered the prototypical behavioral disorder. Pathological gambling did not exist as a separate entity up until the DSM-III, with the early editions of the DSM carrying over the psychoanalytic tradition of perceiving addictions in general, and especially behavioral addictions, as symptoms of deeply rooted character flaws where drugs are employed as artificial defense mechanisms, on the one hand, helping the individual to cope with external and internal stressors, while, on the other hand, leading to a psychological state, comparable with mania, that temporally restores the phase of infantile narcissism [4]. The description of PG, as originally included in the DSM-III, underwent two major revisions up until today. Initially, it was included under the category of “Disorders of impulse control not elsewhere classified”, with an emphasis on the patient’s failure to resist an otherwise unexplained impulse to gamble despite the evidence that it would lead to ruin. The term “behaviour” was introduced in this definition, although it was not critical for the diagnosis of the disorder. The authors still viewed the pathological gambler as an “incomplete” individual who was unable to contain negative affective states, building on more recent advancements in psychoanalytic theory [5]. A change in the perception of PG as a behavioral disorder was evident in the DSM-IV [6], some fourteen years later. What transpired in the meantime, following the seminal work of Mark Griffiths [7], was that the framework of behavioral addictions gradually replaced the framework of hypothesized deficits in impulse control. The set of criteria was revised accordingly, and this revision was tacitly carried over to IAD. Since the majority of IAD cases revolved around pathological online gaming, Internet Gaming Disorder (IGD) was formulated as a separate entity in the DSM-V [8] with a set of criteria mirroring that of PG.

The evolvement of the PG diagnostic entity regarding the impact of bipolar disorder on gambling between the DSM-III and the DSM-IV is also notable. Initially the DSM-III did not explicitly include any phase of the bipolar disorder as an exclusion criterion, listing, instead, antisocial personality disorder. Gambling was also absent as a possible activity manifested in excess during a manic episode, although one of the criteria for a manic episode was “excessive involvement in activities that have a high potential for painful consequences which is not recognized” ([3], p. 209). In the PG section, the DSM-III states: “During a manic or hypomanic episode loss of judgment and excessive gambling may follow the onset of the mood disturbance. When manic-like mood changes occur in Pathological Gambling they typically follow winning” ([3], p. 292). The focus, therefore, in the DSM-III was placed on discerning PG from a manic state rather than their comorbidity.

The DSM-IV revised the exclusion criterion, and, instead of antisocial personality disorder, it featured the wording “the gambling behavior is not better accounted by a manic episode” ([6], p. 618). Obviously, this wording does not exclude a bipolar patient who, at the time, is normothymic, hypomanic, or depressive. This formulation was further elaborated in the text; while excessive gambling may occur during a manic episode, an additional diagnosis of gambling disorder may be given “if the gambling behavior is not better explained by manic episodes (e.g., a history of maladaptive gambling behavior at times other than during a manic episode)” ([6], p. 617), and this exact wording persisted in the DSM-V ([8], p. 589). The DSM-V saw the inclusion of Internet Gaming Disorder as a proposed diagnostic entity, yet no mention was made of a manic episode at all in the description ([8], pp. 795–798) although other mental disorders are referenced as being potentially associated with IGD, including “depressive disorders, attention-deficit/hyperactivity disorder (ADHD), or obsessive-compulsive disorder (OCD)” ([8], p. 797).

Thus, although no mention of a manic episode as a possibility in differential diagnosis is present in the DSM-V, the ICD-11 diagnostic criteria for GD carry over the pathological gambling (ICD-11 code 6C50) exclusion criterion in the peculiar formulation of “The gaming behaviour is not better accounted for by another mental disorder (e.g., Manic Episode).” [1]. Furthermore, the ICD-11 now offers a “boundaries with other disorders and conditions” section, which includes a subsection on “Bipolar and Related Disorders” in which it states: “Increased goal-directed activity including impaired ability to control gaming behaviour can occur during Manic, Mixed, or Hypomanic Episodes. A diagnosis of Gaming Disorder should only be assigned if there is evidence of a persistent pattern of gaming behaviour that meets all diagnostic requirements for the disorder and occurs outside of Mood Episodes” [9].

Again, although not corresponding to the DSM-V set of criteria for IGD but rather to the criteria for PG, even this formulation makes it clear that the existence of bipolar disorder per se is not an exclusion criterion; rather, the expression of gaming addiction symptomatology exclusively during any mood episode, other than a depressive one, would be dismissed as symptoms of a mood swing rather than a co-morbidity. No explanation is offered as to why disordered gaming during a bipolar depressive episode is not considered to be problematic for setting a GD diagnosis as is disordered gaming during a manic, mixed, or hypomanic episode.

Although the precise formulation in the ICD-11, seen above, is much less restrictive than a general exclusion criterion of the mere existence of bipolarity, a question remains as to whether even this formulation is, in fact, evidence-based. This is a crucial issue, since the difficulty of establishing whether a subject with bipolar disorder demonstrates excessive gaming during normothymia leads many researchers to avoid recruiting any bipolar patients in research directed at GD. Thus, to avoid the peril of including a false positive diagnosis of GD in instances where a bipolar patient engages in disordered gaming exclusively during a mood episode other than a depressive one, researchers shun bipolar patients all together. This creates a de facto acknowledgement of causality that is unproven, since if one does not research borderline cases of diagnostic categories, in this case, bipolar patients who demonstrate disordered gaming, one tacitly accepts that these boundaries are valid.

## 2. Materials and Methods

Search strategy

We systematically searched through the electronic databases MEDLINE, Embase, and PsycINFO for published material up until 15 August 2023. The choice of the databases was made according to their focus on medical and psychological research since both entities in question, PG and GD, are typically researched in a medical or psychological context. Additionally, while other social sciences have a valid interest in PG and GD, patient research is not carried out. Search terms were a mixture of bipolar and gaming disorder descriptors including episodes of altered mood. Indicatively, the search terms for MEDLINE were as follows: ((bipolar) OR “bipolar disorder” OR “bipolar affective disorder” OR “manic” OR “mania” OR “bipolar depression”) AND ((internet gaming disorder) OR (gaming disorder) OR (internet addiction) OR (IGD) OR (online gaming)).

For the eligibility criteria, we set the following inclusion criteria: (1) studies should include a cross-examination of gaming disorder and bipolar disorder and not be limited to parallel reporting of incidence, (2) studies focused on either bipolar disorder, GD, or both; (3) studies were journal articles in peer-reviewed publications; (4) studies should have a clearly-defined research population, and (5) studies should describe original research work.

Eighty results were initially recovered; sixty-five were irrelevant, one was relevant but did not offer basic details on the bipolar disorder diagnosis of the patients, and two were reviews. There remained eleven original research articles and a case series. References from these articles were checked to determine any relevant material that eluded the initial search, and a single article was added. The procedures were conducted in accordance with the Preferred Reporting Items for Systematic Reviews and Meta-Analyses (PRISMA) guidelines [10]. The PRISMA flow diagram shows the detailed procedure (Figure 1).

Following the identification of literature, a line was drawn between the studies carried out before and after the official inclusion of GD in the ICD-11 (officially accepted on 24 May 2019, following an initial acceptance procedure dating a year back). The focus on the ICD-11 criteria is due to the fact that GD is the only concisely formulated and accepted version of the disorder, while the IGD diagnosis in the DSM-V did not reference BD at all. This marks a divide between those studies that were supposedly assessed to formulate the ICD-11 criteria (carried out and published before 24 May 2019), versus the studies that were carried out employing these criteria. Publication date is not a reliable marker for this divide since most studies are accepted for publication and made public sometime after their conclusion. Hence, while all studies published before the formulation of the ICD-11 criteria should have been taken into account beforehand, all studies published past the ICD-11 acceptance mark were assessed to determine their actual timeframe. There were only four studies that were carried out and published before the formulation of the ICD-11 criteria [9,11,12,13,14,15]. However, no studies were designed or recruited patients after the formulation of the ICD-11 criteria despite them being published at a later date.

## 3. Results

Table 1 presents an overview of the studies that are reviewed.

### 3.1. Review of the Studies That Were Published before the Formulation of the ICD-11 Criteria

Since these four studies that included bipolar patients were published before the formulation of the ICD-11 criteria for GD, they should have been taken into account during the consultation process. However, a major issue was that all of these early studies did not refer specifically to online (or offline) gaming addiction but rather to the more generic term of IAD. While most heavy internet users, especially adolescents, are typically indulging in online gaming, this is not conclusive. One of the studies that were not included in this review was one of the earliest studies of IAD and its comorbidities by Bernardi and Pallanti that specifically excluded online gaming addicts [24]. Thus, the conclusions reached by studies on internet addiction in general may not necessarily carry over, in their entirety, to gaming addicts. With this in mind, the first study to assess psychiatric comorbidity in IAD was the pioneering study by Shapira et al. [11] in 2000: the authors used a simple definition of “problematic internet use” as (1) uncontrollable, (2) markedly distressing, time-consuming, or resulting in social, occupational, or financial difficulties and (3) not solely present during hypomanic or manic symptoms. Despite the use of an exclusion criterion that was open to interpretation and considerably more austere than that of a full-blown episode of elevated mood, the results were that 70% of all twenty patients met the criteria for a bipolar disorder, while half of them had at least one first- or second-degree relative with a bipolar disorder. An additional 10% (two patients) met the criteria for the bipolar type of schizoaffective disorder. This was both an unpresented finding at the time and a finding that has not been duplicated since. The authors did not mention how many potential patients were, in fact, excluded from the study for demonstrating problematic internet use exclusively during a mood episode, and there was no indication of how many patients were diagnosed with bipolar disorder before their inclusion in the study, although the authors led the reader to presume that bipolarity was not diagnosed prior to the inclusion in the study. That would lead the reader to assume that IAD was the presenting symptom of BD, if, indeed, no other clinically significant events predated the emergence of IAD symptoms. The authors presented their findings as a “preliminary study”, yet no follow-up presentation was ever published.

Park et al. [12], several years later, assessed 795 adolescent Korean students for IAD, depression, suicidal ideation, and bipolarity with four self-report scales. The results were inconclusive because the sample was not large enough, with a small subset of adolescents reporting probable or definite IAD (9.4% or 75 adolescents), of whom four (5.3%) were classified as probably having BD, a frequency that did not differ statistically than those students without IAD, albeit marginally (*p* = 0.059). The authors offered a path diagram in which bipolar symptoms were linked to IAD, yet the inclusion of depressive symptoms in the same path model was odd since they overlapped to a significant extent.

Wölfling et al. [9] screened 368 treatment seekers presenting with excessive to addictive internet use for bipolar spectrum disorders, using a self-report measure (Mood Disorder Questionnaire—MDQ) of thirteen items that assesses the existence of BD. This study was of particular interest since the authors themselves noted the diagnostic issue of comorbidity with BD for IAD (but not specifically GD). They found comorbid bipolar disorders more frequently in patients meeting criteria for internet addiction than among the excessive users (*p* = 0.001). Interestingly, the authors compared bipolar patients with IAD to non-patients with IAD as to their favorite activities; the bipolar group was more often engaged in online pornography and social networking sites. Higher frequency of use for social networking sites was predictive for BD in IAD. Unfortunately, the authors did not specifically mention any results regarding online gaming. However, the combination of a predilection for online pornography and social networking sites may be relatable to a higher interest for sex and meeting a sexual partner, a highlight of manic and hypomanic behavior. A drawback of this study was the use of a single self-report measure of thirteen items to assess the existence of BD. A recent meta-analysis of published studies with the MDQ results indicated adequate validity for patients with prior diagnoses of BD and depression, but it had poor sensitivity in studies of individuals who had not been diagnosed up to that point [25] and were not accustomed to recognizing potential symptoms. Unfortunately, Wölfling et al. [9] did not offer the data as to how many patients had been diagnosed with BD before enrolling in the study and whether any mood episodes had predated the onset of IAD. More importantly, there was no distinction as to whether IAD symptoms were present during elevated mood or during normothymia. Despite not being able to assess the direction of the association between BD and IAD, the authors concluded in a proposal to subsume bipolar disorders as an exclusion criterion for internet addiction without making any distinctions as to the actual online activity which the subject indulged in.

Farahani et al. [13] enrolled 401 Iranian university students in a study assessing the association of IAD and psychiatric disorders by means of self-report tests as well as structured and semi-structured clinical interviews. “Bipolar disorders” were found to increase the odds ratio of IAD by 1,1-fold, which was the lowest of all examined psychiatric disorders except for antisocial personality disorder. Unfortunately, the clinical details of the sample and the details of the statistical analysis are not offered to the reader. The authors do not make it clear whether the term “bipolar disorders” refers to a diagnosis or a number of symptoms. They do not offer any frequencies of disorders in their sample. They do mention that “none of the participants mentioned any mental disorders (including psychotic symptoms (delusion or hallucination)), …hypomania…”. It is unlikely that individuals who do not explicitly mention hypomania or psychotic symptoms at all could be classified as bipolar. Despite employing clinical interviews both for IAD and for psychiatric disorders, the authors do not mention whether symptoms of IAD occurred exclusively during an episode of elevated mood or not.

In total, these four papers, spanning over eighteen years, had serious limitations. Although there is no reason to question the validity of the data presented by Shapira et al. [11], these were not reproducible in the extreme frequency of bipolar patients in IAD-affected individuals, which, even more incredulously, remained undiagnosed up to that point. Park et al. [12] did not conduct an effect-size analysis prior to their study and had an underpowered sample to argue for or against any major point, while their analysis followed a dubious rationale of including self-reported depressive and bipolar symptoms under the same pathway. Farahani et al. [13] presented very little actual data in their paper, to the point of not making it clear whether they were referring to actual diagnoses or a number of symptoms. Wölfling et al. [9] clearly produced the more informative manuscript, since they attempted to directly address the issue, yet they concluded in a suggestion that was not justified by the data at hand, since they methodologically lacked the design to reach a conclusion as broad as they did.

### 3.2. Review of the Studies That Were Published after the Formulation of the ICD-11 Criteria

As mentioned above, despite being published after the formal inclusion of the GD diagnosis in the ICD-11, none of these studies were designed and carried out before that event. Data may have been presented in light of the existence of this diagnosis, however, and thus address some of the diagnostic issues.

Marmet et al. [16] surveyed a large sample of 5516 young Swiss men who enrolled in the military for addictive disorders and psychiatric disorders. Although the researchers focused on alcohol use disorder (AUD), there results were of wider interest since they also included a host of other addictive disorders: cannabis use disorder, tobacco dependence, alcohol use disorder, IAD, GD, smartphone addiction, internet sex addiction, gambling disorder, work addiction, and exercise addiction. Psychiatric disorders included a lifetime diagnosis of BD assessed with the MDQ, major depression, ADHD, and social anxiety. GD in this study was assessed with the Game Addiction Scale [26], a self-report scale that stemmed from the criteria for pathological gambling. The researchers calculated specific predictive models adjusted for sociodemographic variables and other addictions. Participants with AUD and GD were 4.75 times more likely to suffer from major depression, 2.13 times to have ADHD, and 3.06 times to suffer from social anxiety disorder, yet they were not any more likely to have bipolar disorder. Those with AUD and IAD, on the other hand, were 3.13 times more likely to have bipolar disorder.

Di Carlo et al. [17] surveyed a random sample of 772 Italian adults for the co-existence of IAT and psychiatric disorders; manic symptoms as assessed by self-report were more often found in those with an IAD score that pointed to potential addiction. The usefulness of this study for the purpose of this review is limited since it did not explicitly research online gaming habits or establish the timeframe of manic symptoms.

Shao et al. [18] recruited 223 patients from a Chinese center for addictive behaviors from February 2018 to July 2019 to assess whether the diagnosis that gaming addicts receive at a typical visit to a psychiatrist accurately reflects the underlying disorder or not. The authors included only patients who did not have a history or current episode of other primary psychiatric diseases. The authors claim that the clinical diagnosis of GD covered both the diagnostic criteria of the DSM-V and the ICD-11. The results showed that, during the first typical consultation for GD with an untrained psychiatrist, 18.4% of the patients were misdiagnosed with BD, the second most common diagnosis after the one of “mood disorders” (59.2%). Presumably, the authors do not adhere to the classification of BD as a mood disorder. BD was, however, the most common previous diagnosis in the past, with 61% of the patients reporting it, and, subsequently, BD became the most common diagnosis, with 71.2% being diagnosed with BD up to the point of their examination by a trained clinician. The authors therefore draw attention to a critical issue, that of misdiagnosing GD as a symptom of BD. This outcome could arguably be an unintended consequence of the BD exclusion criterion in the latest definitions of GD: a misrepresentation of the exclusion criterion of gaming behavior during elevated mood could lead to overdiagnosis of BD in GD patients. GD patients tend to play with little regard to their personal hygiene, sleep pattern, and eating habits. Their general functioning drops due to their incessant gaming to the detriment of other activities. These features of GD may be perceived as aspects of an episode of BD or, at least, as hypomania.

Carmassi et al. [19] screened 113 subjects with BD for IAD alone with traumatic events and symptoms of post-traumatic stress disorder (PTSD). They found 21.2% of the BD patients had IAD, with those patients reporting a higher number of traumatic events and symptoms of post-traumatic stress. This finding raises the question of whether patients with concurrent BD and IAD have a separate developmental pathway compared to BD patients who do not exhibit IAD. Unfortunately, the authors did not specifically address the particular online activity most frequent in the BD patients who demonstrated IAD.

Ohayon and Roberts [20] examined the prevalence and associations of GD as defined in the DSM-V with BD in a sample of 2984 college students. BD was more often encountered in students positive for GD; the overall prevalence was 6.6% for students with GD versus 1.5% for students in the low risk and 1.4% for students in the moderate risk categories. The authors did not specify whether GD symptoms were present only during a mood episode in BD. Although the difference is statistically significant, the percentage of students with GD and BD was far lower than mentioned in the other studies referenced above [11,18].

Haghighatfard et al. [21] carried out a genome-wide association study of IAD with a host of psychiatric disorders, including BD. They sampled 16,520 patients with IAD, as diagnosed by a clinical interview and a self-report test, and 18,000 matched non-psychiatric subjects. The results indicated that the genetic bases of IAD had the most similarity with autism-spectrum disorders, BD, schizophrenia, and ADHD. The related pathways suggested that deregulations in dopamine release along with dysfunction in CNS development, neuron migration, and neural activity could be liked to IAD. Dopamine pathway abnormalities, in particular, were identified in BD, and this could point to some commonalities between BD and IAD on a basic level. Again, the authors did not specify the particular online activities of IAD-afflicted individuals, and, in this respect, the generalizability of these findings to GD in particular cannot be confirmed.

Borges et al. [22] followed a cohort of 1741 university students in Mexico for a year, following their initial assessment with IGD, which was made with the DSM-V criteria. No long-term associations of mental disorders with baseline IGD were found at the conclusion of the study. This finding shows that IGD is unlikely to serve as a predictor variable for BD.

Floros and Mylona [23], in a case series of GD patients who were treated with a psychoanalytically-informed treatment protocol, presented in detail a case of a young bipolar patient whose GD was a semi-conscious effort to escape a reality where he was limited in his future choices by his BD diagnosis. His online gaming would pick up when his mood was slightly depressed rather than elevated; he could not concentrate enough to be a competitive player during a manic episode or an episode of psychotic depression.

## 4. Discussion

The review of the published material on GD and BD has revealed a number of important facts on the appropriateness of the exclusion criteria for GD.

Up to the point that the ICD-11 criteria for GD were formulated, there were no research studies available that specifically addressed gaming, online or offline, but, rather, there were only a handful of studies that addressed IAD in general. Even after the formulation of the ICD-11 criteria, only three research studies specifically addressed gaming separately from IAD ([16,18,22]). Results from Marmet et al. [16], in fact, showed that IAD and GD had a different relationship to each other with BD. The argument against equating IAD with GD has been supported both by psychological research data [27,28] and by neurophysiological research data [29]. The formulation of the GD exclusion criterion for manic states appears to have been based entirely either on copying over the pathological gambling criteria or on some sort of anecdotal evidence presented during the consultation phase and unknown to the public. Such an approach may have been justified during the formulation of the IGD criteria in the DSM-V, since the IGD diagnosis was under review for inclusion at the time, and those criteria were meant to promote research. This is not, however, acceptable when formulating clear criteria for a disorder which is to be included in the main psychiatric taxonomy. Borges et al. [22] did employ the IGD criteria, and they found no long-term association of the IGD diagnosis with BD a year following diagnosis.There were a number of studies that either lacked essential data on the research population to permit safe conclusions [11,13] or had reached dubious conclusions that were unsupported by the research design [9,12]. Furthermore, the common issue with all the research studies was a failure to distinguish in which phase of BD was addictive gaming present. This is obviously hard to achieve with self-report research instruments unless they include very specific items designed to address this issue. However, this is essential information if we are to reach a conclusive diagnosis in bipolar patients who present with complaints related to potential GD.A very worrying issue was raised by Shao et al. [18], who presented the flip side of the coin in the diagnostic predicament; mental health professionals may attribute GD symptoms to a BD diagnosis. A patient with a behavioral addiction in some instances may resemble a patient in an episode of elevated mood: incessant involvement with gaming with little or no regard for their own well-being and obligations, reduced sleep with disturbance of sleep pattern, reduced need for food or binge-eating, aggressiveness and irritability, and lying or stealing to hide the behavior or to fund in-game purchases. This behavior will appear irrational, and the addicted gamers’ lusting over imaginary characters and fictional tales of fame could be perceived as being detached from reality when taken out of context. In the Shao et al. study, the level of inappropriate BD diagnoses that gaming addicts received was stunning, even more so considering that China has taken serious steps in combatting gaming addiction early on [30] and could be considered to be one of the most sensitized countries to this disorder.No article has appeared in the literature that assesses the frequency of disordered online or offline gaming during a BD episode of any type. One would argue that, if the exclusion criterion is relevant for the GD diagnosis, then there would have been a number of published case reports, case series, and studies in which disordered gaming would be evident and recordable, even quantifiable in some manner. This has not been the case here.

To summarize the above points: the formulation of the exclusion criterion related to BD in the diagnostic criteria for GD in the ICD-11 appears to have been made without sufficient evidence to support its validity. So far, it is unknown if a bipolar patient actually does present a measurable change in his/her gaming habits, whether this occurs in a hypomanic, manic, depressive, or mixed episode, let alone the treatment response regarding this hypothetical symptom. It is worth noting that the definition of BD in the ICD-11 is much more cautious in this respect and does not mention disordered online or offline gaming as a potential marker for any type of BD episode.

These findings also indirectly raise an important concern regarding the validity of the differential diagnosis between PG and GD in general: while both online gambling and gaming are prevalent, the present-day formulations are essentially identical and only refer to a single point of interest: that, in PG, the patient jeopardizes a monetary asset for some sort of gain [1], something that, supposedly, a GD patient does not do. Yet, a GD patient also regularly jeopardizes something of value (educational opportunities, work, and even money in micro-transactions during the game) not only to play but in order to gain something of monetary or emotional value (money through gaming tournaments or advertisements viewed by followers in social media or a sense of gratification achieved through higher ranking and peer recognition). The process of copying over diagnostic criteria from PG to GD may have led to high sensitivity but low specificity in the two diagnoses.

Future research on bipolar patients who engage in problem gambling and gaming could help differentiate the underlying mental processes that occur during gameplay in both disorders by focusing on the differences in frequency, duration, and, most importantly, rationale of gambling and gaming in any phase of the disorder. Bipolar patients who shift through different phases of the disorder thus offer a unique opportunity to research the etiopathology of addictive behaviors, as well as the boundaries between different addictive behaviors.

## 5. Conclusions

This narrative review of the published material on the association of BD and GD has found no reported clinical evidence of disordered gaming during any type of BD episode. The latest definition of GD according to the ICD-11 has copied over the relevant exclusion criterion from the pathological gambling diagnosis with little regard to its actual validity. Most of the studies carried out so far relate to IAD, which is likely to have a different type of association with BD compared to GD. A more relevant issue could be an overdiagnosis of BD in patients with GD due to some overlap in the clinical course of the disorders. More research is required to prove the claim that disordered gaming could be limited in duration and frequency to that of mood episodes in some BD patients. In light of the potential harm of misidentification of GD patients as BD patients, changes to the present set of criteria may be warranted until conclusive evidence is provided. Research in bipolar patients who engage in GD and PG could assist with better differentiation of the two addictive behaviors.

## Figures and Tables

**Figure 1 jcm-12-06251-f001:**
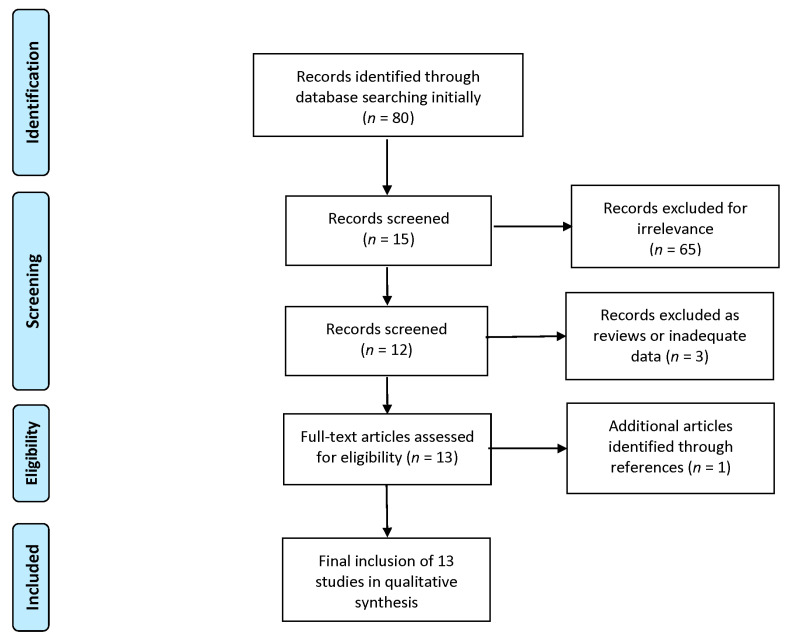
Prisma flowchart of study selection.

**Table 1 jcm-12-06251-t001:** Results from studies carried out on bipolar disorder and gaming disorder.

Reference	Study Design	N	Country	Time-Frame	Age Range	Measures of PG and GD	Findings
Shapira et al. [11]	Descriptive case series	20	US	Study published in 2000	Adults (mean age 36.06 ± 12.0 years)	Structured Clinical Interview for Diagnostic and Statistical Manual of Mental Disorders-IV (SCID-IV), DSM-IV criteria for ICD NOS	A total of 70.0% had a lifetime diagnosis of bipolar disorder (with 12 having bipolar I disorder).
Park et al. [12]	Cross-sectional	795	South Korea	Study published in 2012	Adolescents (mean age, 13.87 ± 1.51 years)	Internet Addiction Proneness Scale for Youth–Short Form (KSscale). The Korean version of the Child Bipolar Questionnaire (CBQ).	A total of 9.4% reported probable or definite IAD, of whom four (5.3%) were classified as probably having BD, a frequency that did not differ statistically from those students without IAD.
Wölfling et al. [9]	Cross-sectional	368	Germany	Data gathered during 2010–2014	Adults (mean age non-BD IAD 24.2 years, mean age BD IAD 26.1 years)	Scale for the Assessment of Internet and Computer Game Addiction (AICA-S) Mood Disorder Questionnaire (MDQ)	Comorbid bipolar disorders more frequent in patients meeting criteria for Internet Addiction than among the excessive users. Higher frequency of use for social networking sites was predictive for BD in IAD.
Farahani et al. [13]	Cross-sectional	401	Iran	Study published in 2018	Adults (age 18–30)	Internet Addiction Test (IAT), Millon Clinical Multiaxial Inventory—Third Edition (MCMI-III), Structured Clinical Interview for DSM (SCID-I), and semi-structured interview	“Bipolar disorders” were found to increase the odds ratio of IAD by 1,1-fold, which was the lowest of all examined psychiatric disorders except for antisocial personality disorder.
Marmet et al. [16]	Cohort, three waves	5516	Switzerland	First wave 08/2010–11/2014. Third wave 4/2016–03/2018	Adults; mean age 19.97 ± 1.22 years old at baseline and 25.47 ± 1.26 at wave 3	Compulsive Internet Use Scale (CIUS), Game Addiction Scale, DSM-5 PG criteria	Patients with alcohol use disorder and IAD were 3.13 times more likely to have BD, while those patents with AUD and GD were not more likely to have BD.
Di Carlo et al. [17]	Cross-sectional	772	Italy	Study published in 2021	Adults; mean age 23.3 ± 3.3 years	Internet Addiction Test (IAT), Mini International Neuropsychiatric Interview	Higher severity of PUI associated with manic/psychotic symptoms.
Shao et al. [18]	Descriptive, case series	223	China	Data gathered during 02/2018–07/2019	Adults/adolescents mean age 20.5 ± 5.1 years	Clinical diagnosis with the DSM-5 and ICD-11 criteria for GD	During the first typical consultation for GD with an untrained psychiatrist, 18.4% of the patients were misdiagnosed with BD.
Carmassi et al. [19]	Descriptive, case series	113	Italy	Data gathered during 11/2016–12/2018	Adults, age range 18–60	Structured Clinical Interview for Diagnostic and Statistical Manual of Mental Disorders-V (SCID-V), single item for online activities	A total of 21.2% of the BD patients had IAD, with those patients reporting a higher number of traumatic events and symptoms of post-traumatic stress.
Ohayon and Roberts [20]	Cross-sectional	2984	US	Data gathered during 2007 and during 2015	Adults; mean age 22.9 ± 5.7 years	DSM-V criteria for GD, self-report for BD	BD was more often encountered in students positive for GD.
Haghighatfard et al. [21]	Cross-sectional case-control	16,520 patients with IAD, 18,000 controls	Iran	Data gathered during 2010–2019	Age range 15–61	Young’s InternetAddiction Test (YIAT), gene analysis for psychiatric disorders genes	The genetic bases of IAD had the most similarity with autism-spectrum disorders, BD, schizophrenia, and ADHD
Borges et al. [22]	Cohort, two waves	1741	Mexico	First wave during 2018–2019, second wave during 2019–2020	Adults (college students)	GD scale based on ICD-11 criteria, Composite International Diagnostic Interview Screening Scales (CIDI-SC)	No long-term associations of mental disorders with baseline IGD were found.
Floros and Mylona [23]	Descriptive, case series	2	Greece	2011–2019	Age 23 and 25	DSM-V diagnostic criteria for IGD, clinical diagnosis for mental disorders	GD symptoms increased during depressive phase of BD.

## Data Availability

No new data were created.

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
