# Peer review of "Bipolar Disorder and Gaming Disorder—Compatible or Incompatible Diagnoses?"

_jcm, 2023, doi:10.3390/jcm12196251_

Round 1

Reviewer 1 Report

Dear Authors.  The work presented is interesting and relevant to the field of action on which it is based.

I consider that the introduction makes an interesting journey through the evolution of the concepts over time in the different well-known classifications, but it does not go in depth in their description, definition and characteristics.  On the other hand, it would be interesting if the objective of the study appeared in a much clearer and more detailed way.

I recommend clarifying the reasons for the choice of the databases used and the exclusion of others that would undoubtedly deserve to be included.

In the results section, it would be much more appropriate to present it in the form of tables with the corresponding study categories previously established.  This would greatly facilitate the reading and comparison of the results obtained.

Finally, bearing in mind that this is a review work, it is important not to lose the descriptive character in the discussion and conclusions section, without entering into inferences beyond what is found.

Best regards.

Author Response

Dear Authors.  The work presented is interesting and relevant to the field of action on which it is based.
 Response: Thank you for your kind words and attention to detail while reviewing our manuscript.
1. I consider that the introduction makes an interesting journey through the evolution of the concepts over time in the different well-known classifications, but it does not go in depth in their description, definition and characteristics.  On the other hand, it would be interesting if the objective of the study appeared in a much clearer and more detailed way.
Response: Thank you for your constructive feedback. Expanding the introduction to include all nuances of the definitions of the two concepts would lengthen the section considerably; we have revisited however the section extensively in order to make the objective clearer and its importance more obvious for the reader. 
2. I recommend clarifying the reasons for the choice of the databases used and the exclusion of others that would undoubtedly deserve to be included.
 Response: Thank you for your suggestion, this clarification has been added in the search strategy section.
3. In the results section, it would be much more appropriate to present it in the form of tables with the corresponding study categories previously established.  This would greatly facilitate the reading and comparison of the results obtained.
Response:  Thank you for your constructive feedback. A table containing all included studies (Table 1) has been added in the manuscript.
4. Finally, bearing in mind that this is a review work, it is important not to lose the descriptive character in the discussion and conclusions section, without entering into inferences beyond what is found.
Response: Thank you for your suggestion. The section has been rewritten where inferences were made and their length was shortened or justified more conclusively.

Reviewer 2 Report

Authors should consider providing clarification between PG and GD at the beginning of the paper. This clarification can set the tone and elucidate the review's intention. Additionally, it may enhance the paper's flow by establishing the connection between the ICD and the DSM.

The PRISMA statement lacks information regarding the number of databases used in the review. Furthermore, the term "IAD" is mentioned in the results section without an explanation of its meaning. The inclusion of adolescent studies should be accompanied by contextual information about the age range pertaining to BAD diagnoses and developmental behaviors.

The conclusions align with the authors' apparent goal of raising questions. It would be beneficial if the authors provided suggestions or engaged in a discussion about how to address this identified gap in the literature.

Flow of introduction is hard to follow.

Author Response

1.Authors should consider providing clarification between PG and GD at the beginning of the paper. This clarification can set the tone and elucidate the review's intention. Additionally, it may enhance the paper's flow by establishing the connection between the ICD and the DSM

Response: Thank you for your suggestion, the Introduction section has been rewritten to address your concerns.

2.The PRISMA statement lacks information regarding the number of databases used in the review.

Response: Thank you for your comment. There were three databases used in the review mentioned in the ‘search strategy’ section and their choice was detailed in the revised manuscript.

3.Furthermore, the term "IAD" is mentioned in the results section without an explanation of its meaning.

Response: Thank you for your comment, the abbreviation has been referenced in the revised Introduction section .

4.The inclusion of adolescent studies should be accompanied by contextual information about the age range pertaining to BAD diagnoses and developmental behaviors.

Response: Thank you for your comment, a new Table that includes all age ranges for the included studies has been added in the manuscript.

5.The conclusions align with the authors' apparent goal of raising questions. It would be beneficial if the authors provided suggestions or engaged in a discussion about how to address this identified gap in the literature.

Response: Thank you for your suggestion, additional points and directions have been added in the revised discussion section.

6.Flow of introduction is hard to follow.

Response: Thank you for your suggestion the introduction section has been revisited with multiple edits for clarity.

Round 2

Reviewer 1 Report

Dear Authors. 

I think you have done a good job of adapting to the proposals of the reviewers.  The article is much clearer.

Best regards.